# Comparative Proteomic Analyses within Three Developmental Stages of the Mushroom White *Hypsizygus marmoreus*

**DOI:** 10.3390/jof9020225

**Published:** 2023-02-08

**Authors:** Lili Xu, Rongmei Lin, Xiaohang Li, Chenxiao Zhang, Xiuqing Yang, Lizhong Guo, Hao Yu, Xia Gao, Chunhui Hu

**Affiliations:** 1Shandong Provincial Key Laboratory of Applied Mycology, School of Life Sciences, Qingdao Agricultural University, 700 Changcheng Road, Qingdao 266109, China; 2College of Plant Protection, Henan Agricultural University, Zhengzhou 450002, China; 3Shandong Agricultural Technology Extending Station, Jinan 250100, China

**Keywords:** *Hypsizygus marmoreus*, proteomics, development, edible mushroom

## Abstract

(1) Background: The *Hypsizygus marmoreus* is a popular edible mushroom in East Asian markets. In a previous study, we reported the proteomic analyses of different developmental stages of *H. marmoreus*, from primordium to mature fruiting body. However, the growth and protein expression changes from scratching to primordium are unclear. (2) Methods: A label-free LC-MS/MS quantitative proteomic analysis technique was adopted to obtain the protein expression profiles of three groups of samples collected in different growth stages from scratching to the tenth day after scratching. The Pearson’s correlation coefficient analysis and principal component analysis were performed to reveal the correlation among samples. The differentially expressed proteins (DEPs) were organized. Gene Ontology (GO) analysis was performed to divide the DEPs into different metabolic processes and pathways. (3) Results: From the 3rd day to the 10th day after scratching, mycelium recovered gradually and formed primordia. Compared with the Rec stage, 218 highly expressed proteins were identified in the Knot stage. Compared with the Pri stage, 217 highly expressed proteins were identified in the Rec stage. Compared with the Pri stage, 53 highly expressed proteins were identified in the Knot stage. A variety of the same highly expressed proteins were identified in these three developmental stages, including: glutathione S-transferase, acetyltransferase, importin, dehydrogenase, heat-shock proteins, ribosomal proteins, methyltransferase, etc. The key pathways in the development of *H. marmoreus* are metabolic process, catabolic process, oxidoreductase activity and hydrolase activity. DEPs in the Knot or Pri stages compared with the Rec stage were significantly decreased in the metabolic-, catabolic- and carbohydrate-related process; and the oxidoreductase, peptidase, and hydrolase activity, which can serve as targets for selectable molecular breeding in *H. marmoreus*. A total of 2000 proteins were classified into eight different modules by WGCNA, wherein 490 proteins were classified into the turquoise module. (4) Conclusions: Generally, from the 3rd day to the 10th day after scratching, mycelium recovered gradually and formed primordia. Importin, dehydrogenase, heat-shock proteins, ribosomal proteins, transferases were all highly expressed in these three developmental stages. DEPs in the Rec stage compared with the Knot or Pri stages were significantly enriched in the metabolic-, catabolic- and carbohydrate-related process; and in oxidoreductase, peptidase and hydrolase activities. This research contributes to the understanding of the mechanisms of the development changes before primordium of *H. marmoreus*.

## 1. Introduction

*Hypsizygus marmoreus* (Peck) H. E. Bigelow, with rich medicinal and high nutritional value (unique seafood flavor and chewy texture), is a widely cultivated commercialized edible mushroom in East Asia and has certain developmental prospects and market potential due to the obvious market demand and price advantages and the small planting scale [1,2,3,4]. A wide variety of components with different biological activities have been identified in *H. marmoreus*; for instance, protocatechuic acid, gallic acid, polysaccharides, and (+)-catechin [4]. These compounds and their derivatives play significant roles in anti-inflammation, anti-oxidation, promising anti-melanoma effects, inhibiting the growth of human myeloid leukemia U937 cells, immunomodulatory activities, and antihypertensive action [5,6]. However, the growth cycle of *H. marmoreus* is long compared with other industrially cultivated mushrooms, such as *Agaricus bisporus*, *Pleurotus eryngii*, and *Flammulina velutipes*. It is necessary to improve the growth efficiency of this mushroom. Understanding the molecular mechanisms underlining *H. marmoreus* development is necessary for the genetic breeding of this mushroom.

The growth and development of *H. marmoreus* can be divided into three stages, the vegetative growth stage, the transition stage, and the fruiting stage (reproduction growth). The transition stage from vegetative growth to fruiting of *H. marmoreus* contains the following steps: mycelial scratching and watering (with changed environmental factors), recovery growth stage, mycelial knotting stage, and primordium stage (followed by the nail head stage). After maturation of the vegetative growth of the mycelia, the cap of the bottle was removed to increase the oxygen supplementation and stimulate the mycelia directly with lower temperatures and light stimulation [7,8,9,10]. Meanwhile, the port skin on the surface of the culture media was scratched. The purpose of the scratching step is to stimulate the formation of the fruiting body by making a mechanical wound, making the mycelia develop simultaneously, and the flat mushroom-producing surface is conductive to the consistency of mushroom production, which can increase the quality of the fruiting body [11,12,13]. After scratching, water was added to the bottle and absorbed by the culture, activating the metabolic activity of the after-ripening mycelia [14,15]. Meanwhile, the cultivation temperature was changed from ~25 to 16–18 °C with a humidity of 95–99%. The growth recovery of the mycelia takes ~5 days. [16,17]. After the recovery stage, the temperature was changed to 12–14 °C with a humidity ≥95% and longer periods of light to stimulate the development of the recovered mycelia [18,19]. Under such conditions, the mycelia twisted in this period, and then the primordia could be found on the scratching surface for about 7 to 9 days, resulting in the formation of different sizes of gray or yellowish-brown mushroom buds [20]. Once the primordia formed, the temperature was adjusted to 15–17 with a humidity of 95–98%, a CO_2_ concentration of 1000–1500 rpm, and light conditions. After 9–10 days the fruiting body developed to harvest maturity [21,22]. In the transition stage, the mycelia experienced drastic environmental changes such as water, temperature, humidity, light, and CO_2_ concentration changes. Research into the protein changes in this period is crucial for understanding *H. marmoreus* fruiting body development. However, little information has been reported on the gene and protein expression in this period.

In a previous study, we reported the proteomic analyses of different developmental stages during the cultivation of *H. marmoreus*: the mycelium stage, the primordia stage, the immature fruiting bodies stage (small and large), and the mature fruiting bodies stage (cap and stem) [22]. Samples for proteomic analyses were taken from the thirteenth day after scratching in our previous study. However, the development and protein changes in the transition stage are unclear [1,23]. In this research, proteomic analysis of three stages of *H. marmoreus* from scratching to primordium were carried out, which supplemented data of the developmental stages to our previous research. These three stages include the Rec (recovery-cultured mycelium: on the 3rd day after scratching), Knot (mycelium kink stage, on the 7th day after scratching) and Pri (the primordia just formed on the 10th day after scratching) stages.

## 2. Materials and Methods

### 2.1. Culture Conditions and Acquisition of the H. marmoreus Samples

*H. marmoreus* strain G12 was provided by the Laboratory of Mushroom Precision Breeding at Qingdao Agricultural University (mushroomlab.cn (accessed on 1 December 2022)). The mycelia of this strain were maintained and cultivated as previously described [22]. The fruiting process was carried out in a mushroom factory in Dongying, Shandong Province, China. The spawn was inoculated in a bottle-cultured medium and then cultivated at 25 °C for 80–90 days. After that, the surface mycelia were scratched and water was replenished into the cultivation bottle, and the date was set as day 0. The fungal samples were collected on the 3rd day after scratching (recovery-cultured mycelium, Rec group), on the 7th day after scratching (mycelium knotting stage, Knot group), and on the 10th day after scratching (primordia just formed, Pri group). Three biological repeats for each stage were snap-frozen in liquid nitrogen and stored at −80 °C until proteomic analysis.

### 2.2. Protein Extraction and Peptide Digestion

Total proteins were extracted from the frozen *H. marmoreus* samples according to the following protocol. First, 100 mg of frozen sample was placed into centrifuge tubes and then 1 mL UT buffer (8 M urea, 0.1 M Tris-HCl, pH 8.5) containing HaltTM Protease inhibitor Cocktail was added. The tissueLyser II was used to break the sample at 150 Hz for 60 s. The cell extract was treated by ultrasonication for 24 s (on for 6 s, off for 15 s). Tissue debris was removed by centrifugation (12,000× *g* for 10 min at 4 °C), and the supernatant was transferred to new tubes. The protein concentration was determined using the Bradford Protein Assay Kit (Thermo Fisher, Waltham, MA, USA). The proteins were reduced by adding 15 mg DTT and incubated at 37 °C for 1 h [23,24].

Enzymolysis was performed according to the FASP method created by Wisniewski et al. [25]. The extracted proteins (100 μg) were dissolved in 300 μL UT buffer and placed into Pierce Protein Concentrators PES (10 K MWCO, 0.5 mL) (Thermo Fisher) to remove low molecular impurities by centrifugation at 10,000× *g* for 30 min. 50 mM of iodoacetamide was added to alkylate the proteins for 30 min at room temperature in the dark. The proteins were washed with 200 μL UT and 300 μL of 50 mM NH_4_HCO_3_ after removing the buffer by centrifugation. 2 μg of modified trypsin (Thermo Fisher) in 100 μL of 50 mM NH_4_HCO_3_ was added to the ultrafiltration tube in a mass proportion of 1:50 (trypsin/protein). Enzymolysis was performed with gentle shaking at 37 °C for 12 h. After this, peptides were collected by centrifugation at 10,000× *g* for 15 min, and the residue peptides in the ultrafiltration tube were washed with 50 μL 50 mM NH_4_HCO_3_ one more time. The salt of the pooled elutes was removed by using C18 SPE (Empore). Additionally, the peptide concentration was measured by utilizing the Pierce Quantitative Colorimetric Peptide Assay. Subsequently, the peptide samples were lyophilized on an RVC 2-25 CD plus vacuum concentrator (Christ), and stored at −80 °C for further analysis.

### 2.3. Label-Free LC-MS/MS Quantitative Proteomic Analysis

This analysis was performed by Novogene Co., Ltd. (Beijing, China). The desalted peptides were reconstituted in 10 μL 0.1% formic acid. LC-MS analysis was performed on a Nano-LC system coupled with an Orbitrap Fusion^TM^ Tribrid^TM^ (Thermo Fisher Scientific). The MS instrument was operated in the data-dependent acquisition (DDA) mode, with full MS scans over a mass range of 350–1500 *m*/*z* with detection in the Orbitrap (120 K resolution) and with auto gain control set to 100,000. Different chromatographic gradient lengths from 60 to 240 min were tested for peptide separation. All gradients started at 5% *v*/*v*) acetonitrile (ACN) (0.1% formic acid) and went up to 32% (*v*/*v*) (ACN (0.1% formic acid). Three biological repeats for each group were performed [26]. 

### 2.4. Peptide and Protein Identification

The raw data were analyzed using Proteome Discoverer software suite version 2.0 (Thermo Fisher Scientific) against the *H. marmoreus* protein database from Uniprot (UP000076154) [27]. Protein identification was supported by at least two unique peptides with a false discovery rate lower than 0.05.

### 2.5. Bioinformatic Analysis

Raw data obtained from the Proteome Discovery software were normalized. The *H. marmoreus* proteome was annotated and functionally enriched using the Gene Ontology tool (http://geneontology.org/, (accessed on 1 December 2022)) according to cellular components (CCs), molecular functions (MFs), and biological processes (BPs). A weighted gene co-expression network analysis (WGCNA) was used to interpret gene expression data [28]. GO analysis was visualized using the R-ggcyto tool [29]. 

## 3. Results

### 3.1. Phenotype of H. marmoreus in Three Developmental Stages from Scratching to the Thirteenth Day after Scratching

To investigate the development and protein changes from scratching to the primordia stage of *H. marmoreus*, mycelium samples on the 3rd day (Rec group), 7th day (Knot group) and 10th day (Pri group) after scratching were collected (Figure 1). After the old mycelium skin was scraped off, the mycelia on the culture surface gradually recovered. On the 3rd day after scratching, white mycelium without differentiation emerged, and the recovery mycelia covered almost the entire surface of the culture (Figure 1A). On the 7th day after scratching, the mycelium knotting stage, the thick, separated mycelia disappeared, and mycelia twisted together to form white, granular, dense kinking tissues (Figure 1B). On the 10th day after scratching, the primordia begin to form and some primordia showed signs of bud differentiation (Figure 1C). In general, from the 3rd to the 10th day after scratching, mycelium transformed from vegetative to reproductive growth.

### 3.2. Quantitative Visualization of the Differential Expression of Proteins of H. marmoreus in Three Developmental Stages 

Principal component analysis showed that the three replicates of the Pri (green) groupwere more closely related than the Knot (red) and Rec (purple) groups. Compared with the Pri group, the Knot and Rec group demonstrated a more divergent expression level. The first principal component varied in Knot, while the second principal component varied in the Rec group (Figure 2B). Pearson’s correlation analysis showed that the correlation coefficient among the different treatments varied from 0.75 to 1 (Figure 2A,C, Appendix A). The protein expression profiles of the Pri and Knot groups displayed a higher correlation than the correlation between the Pri and Rec groups, consistent with the principal component analysis (Figure 2A–C). Additionally, Pearson’s correlation coefficient analysis and principal component analysis results revealed that the three groups of samples were reasonable with good correlations among the biological repeats.

Compared with *H. marmoreus* in the Knot stage, there were 220 highly expressed proteins in the Rec stage in *H. marmoreus* (Figure 3, Appendix A). A total of 32 out of 220 DEPs were uncharacterized proteins, which remains to be annotated. DEPs include carboxypeptidases (e.g., A0A369JGP6, A0A369K1C4, and A0A369K147), glutathione S-transferases (e.g., A0A369JPN9, and A0A369JYR2), cuticle-degrading proteases (e.g., A0A369KCM9, and A0A369JNB4), alcohol dehydrogenases (e.g., A0A369JW11, A0A369J8X4, and A0A369JQM8), and hydrolases (e.g., A0A369JIP2, and A0A369JZK0). Compared with *H. marmoreus* in the Pri stage, there were 217 highly expressed proteins in *H. marmoreus* in the Rec stage (Figure 4, Appendix A). These DEPs included carboxypeptidases (e.g., A0A369JGP6, A0A369K1C4, and A0A369K147), dehydrogenases (e.g., A0A369K8H0, and A0A369KBH4), acetyltransferases (e.g., A0A369K040, A0A369IYD4, and A0A369K1T5), peroxidases (e.g., A0A369JNT7, and A0A369JL11), and chitinases (e.g., A0A369JWX5, A0A369JFV7, and A0A369JFT6). Compared with *H. marmoreus* in the Pri stage, there were 53 highly expressed proteins in *H. marmoreus* in the Knot stage (Figure 5, Appendix A). The DEPs included binding proteins (e.g., ATP, NAD, and FMN), carbohydrate, calcium ion-binding proteins (e.g., A0A369J8I9, A0A369K6W1, A0A369J580, and A0A369K2E9), hydrolases (e.g., A0A369JNH4, and A0A369JF95), peroxidases (e.g., A0A369JID4, and A0A369JL11), oxidoreductases (e.g., A0A369JZ61, and A0A369K8F2) and ribosomal proteins (e.g., A0A369JPZ9, and A0A369JT05).

Proteins highly expressed in the Pri stage compared with the Rec stage in *H. marmoreus* included: ribosomal proteins, dehydrogenase, methyltransferase, diphosphate synthase, proteasome, acetyltransferase, kinase, translation initiation factor, glutathione S-transferase, heat-shock proteins, importin, and mitochondrial import inner membrane translocase (Figure 6A). Proteins highly expressed in the Knot stage compared with the Rec stage in *H. marmoreus* included: ribosomal proteins, diphosphate synthase, dehydrogenase, methyltransferase, proteasome, acetyltransferase, kinase, translation initiation factor, glutathione S-transferase, heat-shock proteins, importin, peroxiredoxin, hydrolase, oxygenase, ABC transporter, decarboxylase, and carboxykinase (Figure 6B). Proteins highly expressed in the Pri stage compared with the Knot stage in *H. marmoreus* included: ribosomal proteins, ATP synthase, cAMP-dependent protein kinase, glutathione S-transferase, heat-shock proteins, mitochondrial import inner membrane translocase, NADH dehydrogenase, UDP-glucose pyrophosphorylase, serine protease inhibitor, elongation factor, cytochrome oxidase, reductase, importin, methyltransferase, acetyltransferase, acyl-CoA-binding protein, peroxiredoxin, nuclease, and oxidoreductase, (Figure 6C).

### 3.3. Gene Ontology (GO) Enrichment Analysis for DEPs

GO analysis for the DEPs to molecular functions (MFs), cellular components (CCs), and biological processes (BPs) categories are as follows. Gene Ontology analysis was performed to classify the annotated DEPs. These profiled DEPs were categorized into three main GO categories: CCs, MFs, and BPs.

In total 220 proteins were highly expressed in the Rec stage compared with the Knot stage in *H. marmoreus* (Figure 3). These 220 DEPs were divided into 32 terms, involving 10 BP, 3 CC, and 19 MF terms. The significantly enriched terms of each category were as follows: carbohydrate metabolic process (16), catabolic process (13), organic substance catabolic process (12), cellular catabolic process (9), organonitrogen compound catabolic process (6), and macromolecule catabolic process (6) for BPs; organelle membrane (7), catalytic complex (5), and mitochondrion (5) for CCs; and catalytic activity (126), hydrolase activity (100), oxidoreductase activity (42), peptidase activity (21), serine-type peptidase activity (10), carbohydrate binding (8), and endopeptidase activity (8) for MFs (Figure 7 and Figure 11, Appendix A). These DEPs may be involved in the recovery of cultured *H. marmoreus* mycelium.

The carbohydrate metabolic processes, catabolic processes, and other metabolic processes are active in the recovery-cultured mycelium stage, which indicates that the cellular metabolic process is active on the 3rd day after scratching. From the molecular functioning aspect, hydrolase, catalytic, and peptidase activity, as well as oxidoreductase also confirm that material transformation and metabolism occur frequently in *H. marmoreus* on the 3rd day after scratching. 

In total 217 proteins were highly expressed in the Rec stage compared with the Pri stage in *H. marmoreus* (Figure 4). These 217 DEPs were divided into 29 terms, involving 12 BP, 1 CC, and 16 MF terms. The significantly enriched terms of each category included: catalytic activity (130), hydrolase activity (141), oxidoreductase activity (37), and peptidase activity (35) for MFs; extracellular region (6) for CCs; metabolic process (116), macromolecule catabolic process (10), polysaccharide catabolic process (9), carbohydrate catabolic process (9), cellular catabolic process (10), etc., for BPs (Figure 8 and Figure 11, Appendix A). These DEPs may be involved in the recovery of cultured *H. marmoreus* mycelium. Extracellular region and polysaccharide catabolic process was significantly enriched in the Rec stage compared with the Pri stage, but not enriched in the Rec stage compared with the Knot stage.

The metabolic process and catabolic process are more active on the 3rd day after scratching than on the 7th or 10th day after scratching, which suggest that scratching is a physical stimulation to mycelium so that metabolic, catabolic, and material transformation occur frequently in the first few days after scratching. Scratching is to use the machine to cut off the hyphae after finishing the maturation to induce fruiting and to create a flat surface to ensure orderly fruiting. When the primordia forms, all these metabolic and transformation processes tend to be slower. 

A total of 53 proteins were highly expressed in the Knot stage compared with the Pri stage in *H. marmoreus* (Figure 5). These 53 DEPs were divided into 21 terms, involving 17 BP and 4 MF terms. The significantly enriched terms of each category were as follows: catalytic activity (30), oxidoreductase activity (10), and hydrolase activity (10) for MFs; organic substance metabolic process (22), primary metabolic process (20), metabolic process (22), biosynthetic process (12), organonitrogen compound metabolic process (11), oxoacid metabolic process (6), cellular amide metabolic process (6), and catabolic process (5) for BPs (Figure 9 and Figure 11, Appendix A). According to the GO terms, the primary and cellular amide metabolic processes were more active in *H. marmoreus* on the 7th day rather than the 10th day after scratching. This is because in the several days from scratching to primordia, basic metabolic and basic life process are active and primary metabolite, such as amino acid, polysaccharides, lipids, nucleotides, and vitamins, are produce in the middle period of days from scratching to primordia (on the 7th day).

### 3.4. Weighted Gene Co-Expression Network Analysis (WGCNA) for DEPs

2000 proteins were classified into eight different modules by the WGCNA (Figure 10B). A total of 490 proteins were classified into a turquoise module, which is the largest module. Ninety-six proteins were classified into the black module. A total of 484 proteins were classified into the blue module. A total of 397 proteins were classified into the brown module. A total of 137 proteins were classified into the green module. A total of 23 proteins were classified into the grey module. A total of 53 proteins were classified into the pink module. A total of 135 proteins were classified into the red module. A total of 185 proteins were classified into the yellow module. The eigengene of each module in each treatment shows the different expression levels of genes in each module. Proteins in the MEbrown module showed relatively high expression levels in the Knot and Pri groups compared with the Rec group, whereas proteins in the MEpink module showed relatively high expression levels in the Rec and Knot groups compared with the Pri group, proteins in the MEyellow module displayed relative high expression levels in the Pri and Rec groups compared with the Knot group (Table 1, Figure 10). The eigengene adjacency heatmap shows that MEred and MEgreen cluster with MEturquoise and MEpink, and MEyellow clusters with MEbrown, MEblack, and MEblue (Figure 10A). There were 20 hub genes for each module. Many modules showed more or less intersection in gene function.

According to different expression patterns the MEpink, MEyellow, and MEbrown modules were further analyzed. In the MEpink module, the hub genes function as lactoylglutathione lyase, kynurenine formamidase, malic enzyme, GTPase, peroxidase, ornithine aminotransferase, hydroxy-nicotine oxidase, GTP cyclohydrolase, phosphopyruvate hydratase, oxidoreductase, etc. (Figure 11). These proteins are involved in the mycelia recovery and mycelium kink stages. In the MEyellow module, the hub genes function as exoribonuclease, fumarate reductase, glutaredoxin, mRNA-splicing factor, hydroxyethylthiazole kinase, leucine zipper, oxidoreductase, aldehyde dehydrogenase, RNA helicase, glucosidase, translation initiation factor, peptidylprolyl isomerase, fructose-bisphosphate aldolase, phosphatidylinositol transfer protein, etc. These proteins are associated with recovery mycelium and primordia formation stages (Figure 10 and Figure 11, Appendix A). In the MEbrown module, the hub genes function as ribosomal protein, heat-shock protein, elongation factor, dehydrogenase, proliferation-associated protein, phosphofructokinase, methyltransferase, isoprenyl diphosphate synthase, ligase, kinase, etc. (Figure 11). These proteins are associated with the mycelium kink and primordia formation stages.

## 4. Discussion

In a previous study, proteome analyses of different developmental stages during the cultivation of *H. marmoreus* were carried out in the mycelium, the tissues from primordia to mature fruiting body [26]. Samples were taken from the thirteenth day after scratching in our previous study. Very few studies report the changes in expressed proteins in the multiple successive developmental stages in edible mushrooms. However, until this research, the development and proteome changes from scratching to the thirteenth day after scratching were unclear. The whole genome of *H. marmoreus* has been assembled and annotated in high quality. In this research, proteome analysis of three developmental stages of *H. marmoreus* from scratching to the thirteenth day after scratching was carried out, supplementing data of the developmental stages from our previous research. These three stages include Rec (recovery-cultured mycelium: on the 3rd day after scratching), Knot (mycelium kink stage, on the 7th day after scratching), and Pri (the primordia just formed, on the 10th day after scratching). We found that, from scratching to primordia formation, metabolic, catalytic, and component transformation processes become gradually slower. The differentiation and development from the 3rd day to the 10th day is a continuous process; therefore, a variety of the similar highly expressed proteins were identified in these three developmental stages from the 3rd day to the 10th day after scratching, including glutathione S-transferase, acetyltransferase, importin, dehydrogenase, heat shock proteins, ribosomal proteins, methyltransferase, etc. 

Glutathione S-transferase is secreted/sequestrated and conjugated achieving intracellular detoxification. Three subfamilies of the fungal GST family expand in wood-decaying fungi: the GST Omega, the GSTFuA class, and the GST Ure2p. The omega class of glutathione S-transferase expands in the order of Boletales and Polyporales [30]. The multiplication of the omega class of glutathione S-transferase allows mushrooms to adapt to new environments more easily. From the 3rd day to the 10th day after scratching, glutathione S-transferase is necessary for intracellular detoxification after scratching. Acetyltransferase protects cells from damage caused by ROS (decreasing reactive oxygen species) in oxidative pressure environments; for example, heat-shock, freeze–thawing or ethanol treatment in *Saccharomyces cerevisiae* [31]. Therefore, acetyltransferase is indispensable to protect cells from damage caused by ROS and recovery due to damage caused by scratching from the 3rd day to the 10th day after scratching.

Heat-shock proteins are highly conserved proteins in all organisms, which regulate organism growth and development, as well as response to environmental stress. What is more, heat-shock proteins are also identified as playing an indispensable role in the development process of mushroom. The function of heat-shock proteins in response to different types of stress has been proven in mushrooms. There are eight heat-shock proteins after scratching in *H. marmoreus* (i.e., A0A369JS72, A0A369JLF5, A0A369JNN0, A0A369JUF6, A0A369K2K3, A0A369JVD2, A0A369JBY3, and A0A369KH52). Heat-shock proteins were highly expressed proteins identified in these three developmental stages from the 3rd day to the 10th day after scratching. Mitochondrial protein translocases, particularly the subunits of the ATOM complex, are involved in the import of tRNA in organisms [32]. Liu et al. reported that heat-shock proteins are involved in the development of *F*. *velutipes* and the same results were also reported in our previous research on *H. marmoreus* fruiting body development. The results indicated that the stress response system plays an important role in the development of the fruiting body of the mushroom [33]. 

Mitochondrial import inner membrane translocase was highly expressed in the Pri stage compared with the Knot and Rec stages. UDP-glucose pyrophosphorylase (UGP: UDP-xylose-4-epimerase and UDP-glucose-4-epimerase) plays a significant role in mycelial growth and polysaccharide synthesis in the development of mushrooms [34]. UDP-glucose pyrophosphorylase was highly expressed in the Pri stage compared with the Knot stage. The translation initiation factor was highly expressed in the Pri and Knot stages compared with the Rec stage, which could serve as a potential stage-specific biomarker to research the fruiting process in mushrooms [34]. Enzymes for steroid transformations in mushrooms can be divided into two groups: (i) steroid-transforming enzymes, for instance, oxidoreductases (3β-hydroxysteroid dehydrogenase/isomerase, 5α-reductase), a wide variety of different hydroxylases (7α-, 11α-, 11β-, 14α-hydroxylase), 17β-hydroxysteroid dehydrogenase. (ii) Enzymes belonging to the ergosterol biosynthetic pathway, such as 14α-demethylase, C-24 methyltransferase, C-22 desaturase and C-24 reductase [35]. Dehydrogenase and methyltransferase were also highly expressed proteins identified in these three developmental stages from the 3rd day to the 10th day after scratching. Oxidoreductases were highly expressed in the Pri stage compared with the Knot stage in *H. marmoreus* since when the primordia just formed, with frequent material reductions. Ribosomal proteins from *Lentinula edodes* and *Pleurotus eryngii* mushrooms are cross-reactive. Ribosomal proteins of fungi, such as *Alternaria alternata* and *Aspergillus fumigatus,* are supposedto mold allergens. These fungi usually cause respiratory allergic diseases. Food allergies similar to button mushroom (*Agaricus bisporus*) or mycoproteins due to cross-reactive with molds have been researched [32,33,36,37]. Ribosomal proteins are both highly expressed proteins identified in these three developmental stages from the 3rd day to the 10th day after scratching.

Filamentation of mushrooms is mediated by second messengers such as cAMP (cyclic adenosine 3′,5′-monophosphate) synthesized by adenylyl cyclase. Liu et al. reported that the MAPK signaling pathway might be involved in the fruiting body development in *F. velutipes* [38]. The MAPK signaling pathway (Mkk1_2 and PBS2) and cAMP signaling (PKA) pathways are involved in the growth and development of *H. marmoreus*. The cAMP-dependent protein kinase regulatory subunit (A0A369JT35) is highly expressed in the Pri stage compared with the Knot stage in *H. marmoreus.* Zinc finger transcription factor YRR1 (A0A369K815) and the general negative regulator of transcription subunit (A0A369JKD5, and A0A369J2L6) are involved in mycelium development after scratching. The results suggested that these regulators or signal transporters are involved in the transition from vegetative to reproductive growth in *H. marmoreus*.

Several interesting proteins are potentially associated with mycelium development after scratching, including hydrolase, esterase, oxidoreductase, dehydrogenase, and so on. Functional analysis of these proteins might contribute to revealing some novel aspects of known and important processes and provide new strategies for improving the cultivation of edible mushrooms [6]. In total 60 hydrolases, 11 esterases, 32 oxidoreductases, and 123 dehydrogenases were identified as differentially expressed proteins from the 3rd to the 10th day after scratching in *H. marmoreus*. Oxidoreductases were highly expressed in the Pri stage compared with the Knot stage. Oxidoreductases, such as tyrosinase, play significant roles during the catalytic reaction of melanin in the development of mushrooms. Hydrolases were highly expressed in the Knot stage compared with the Rec stage. These results indicated that oxidoreductases are involved in the transition from vegetative to reproductive growth and these oxidoreductases might synthesize important compounds driving the development of *H. marmoreus*. Further study is needed to investigate the function of metabolites in the transition stage of *H. marmoreus* development.

## 5. Conclusions

This study investigated the proteome expression in the transition stage from vegetative to reproductive growth in *H. marmoreus* after scratching. Generally, from the 3rd day to the 10th day after scratching, mycelium recovered gradually, knotted, and formed primordia. The differential expressed proteins in the Rec, Knot, and Pri stages were presented and discussed in the present study. In conclusion, importin, dehydrogenase, heat-shock proteins, ribosomal proteins, and transferases were differentially expressed in these three developmental stages. The key pathways in the development of *H. marmoreus* were metabolic processes, catabolic processes, oxidoreductase activity, and hydrolase activity, which could serve as targets for the selectable molecular breeding in *H. marmoreus*. This research contributes to understanding the developmental mechanisms of the before the primordia of *H. marmoreus*. Further research is still needed to investigate the function of other key proteins.

## Figures and Tables

**Figure 1 jof-09-00225-f001:**
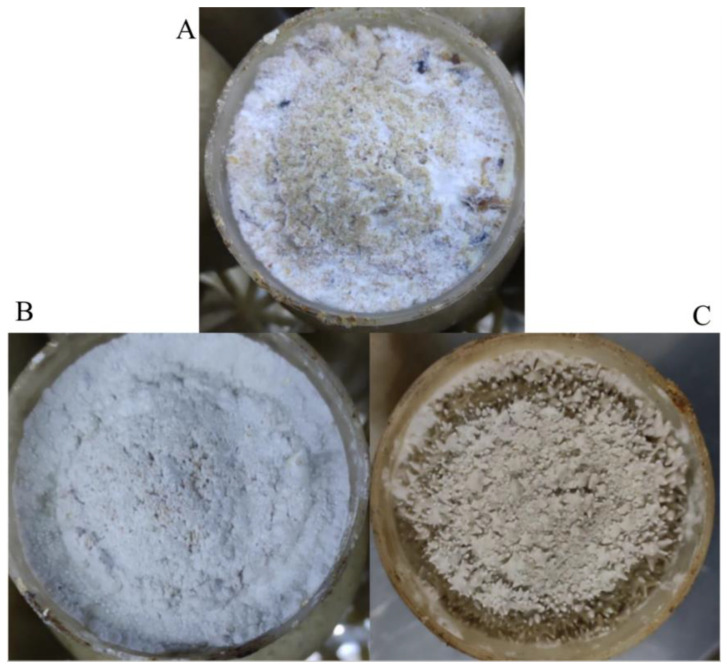
Morphological comparison of three developmental stages of *H. marmoreus* from scratching to the thirteenth day after scratching. (**A**) Rec (recovery-cultured mycelium, on the 3rd day after scratching). (**B**) Knot (mycelium kink stage, on the 7th day after scratching). (**C**) Pri (the primordia just formed, on the 10th day after scratching).

**Figure 2 jof-09-00225-f002:**
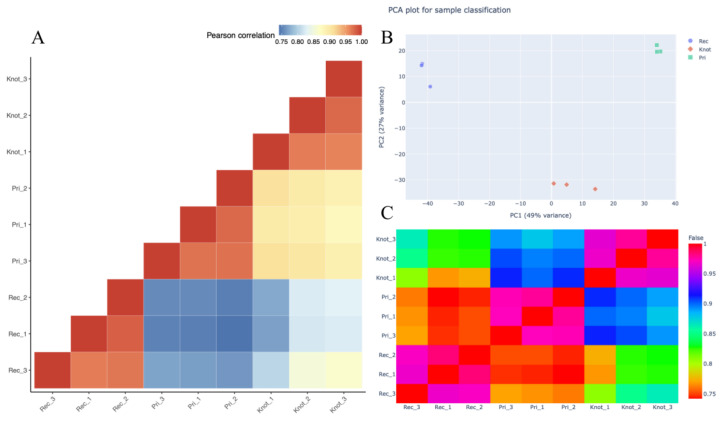
PCC and PC analysis representations of the proteomic dataset of the three groups of samples. (**A**) Pearson’s correlation coefficient analysis for pair-wise comparisons of the proteome data. (**B**) Principal component analysis of the proteome data from the three groups of samples. The Rec, Knot and Pri groups were located separately. (**C**) Pearson’s correlation heatmap results for the pair-wise comparisons of proteome data.

**Figure 3 jof-09-00225-f003:**
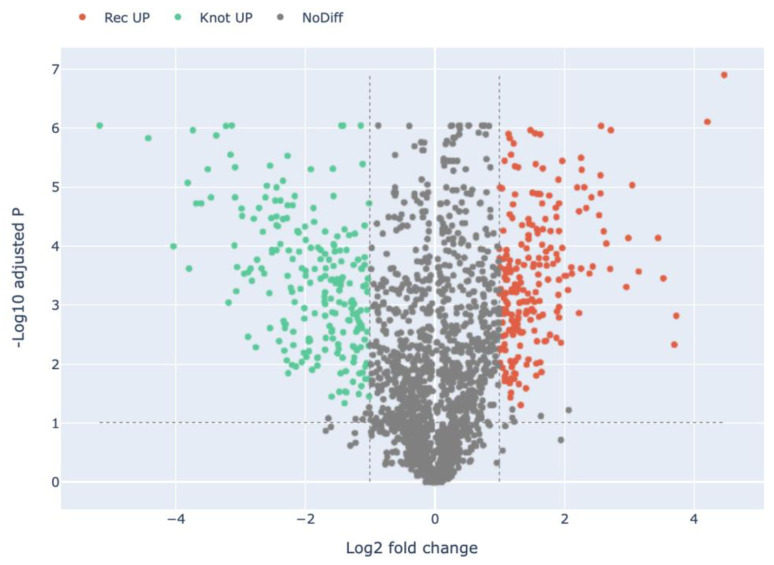
Volcano plot of the differentially expressed proteins in the Rec group compared with the Knot group in *H. marmoreus*. Red dots indicate the up-regulated proteins in the Rec group. Green dots indicate the up-regulated proteins in the Knot group. Grey dots indicate proteins that are not significantly differentially expressed.

**Figure 4 jof-09-00225-f004:**
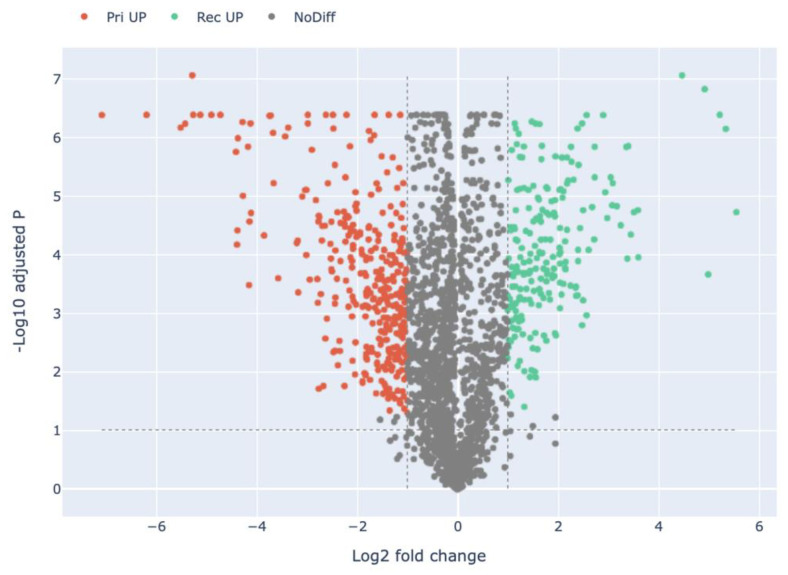
Volcano plot of the differentially expressed proteins in the Pri group compared with the Rec group in *H. marmoreus*. Red dots indicate the up-regulated proteins in the Pri group. Green dots indicate the up-regulated proteins in the Rec group. Grey dots indicate proteins that are not significantly differentially expressed.

**Figure 5 jof-09-00225-f005:**
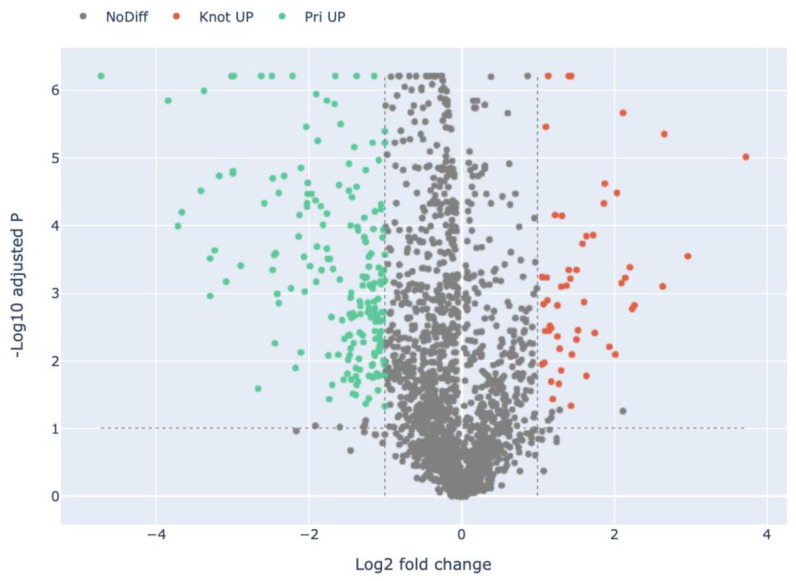
Volcano plot of the differentially expressed proteins in the Knot group compared with the Pri group in *H. marmoreus*. Red dots indicate the up-regulated proteins in the Knot group. Green dots indicate the up-regulated proteins in the Pri group. Grey dots indicate proteins that are not significantly differentially expressed.

**Figure 6 jof-09-00225-f006:**
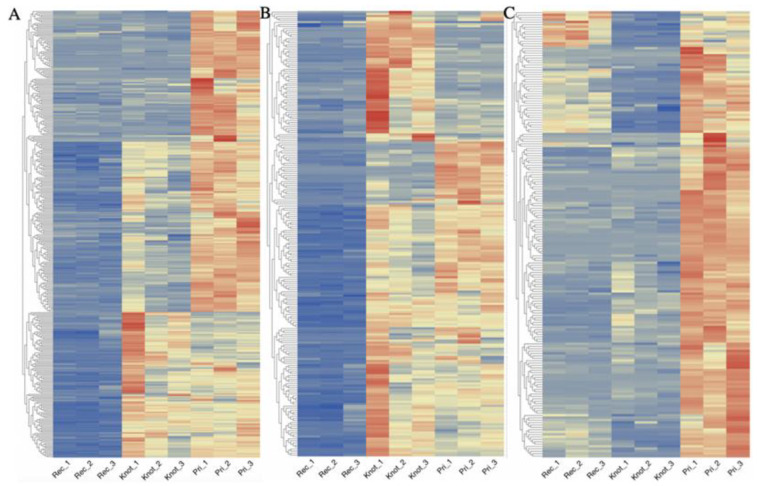
Heatmap of differentially expressed proteins in the three groups. (**A**) Heatmap of proteins highly expressed in the Pri stage compared with the Rec stage in *H. marmoreus*. (**B**) Heatmap of proteins highly expressed in the Knot stage compared with the Rec stage in *H. marmoreus*. (**C**) Heatmap of proteins highly expressed in the Pri stage compared with the Knot stage in *H. marmoreus*.

**Figure 7 jof-09-00225-f007:**
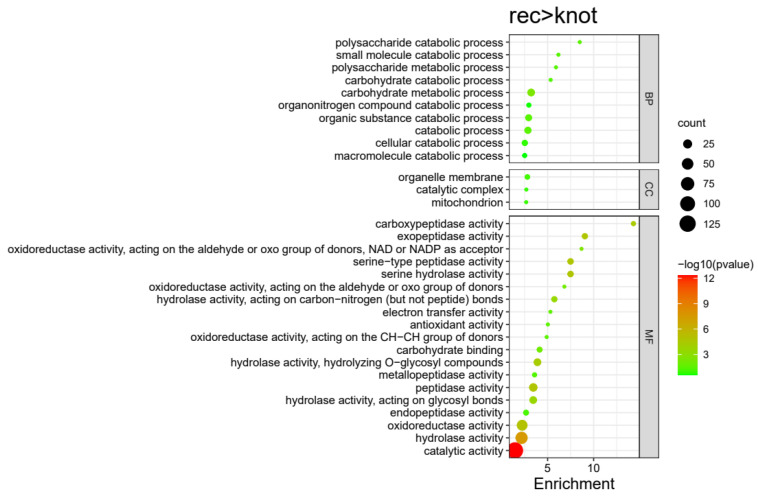
Enriched GO terms of highly expressed proteins in the Rec group compared with the Knot group in *H. marmoreus*. Bubble size indicates protein counts. Bubble color indicates p value. BP represents biological process, CC represents cellular component, and MF represents molecular function.

**Figure 8 jof-09-00225-f008:**
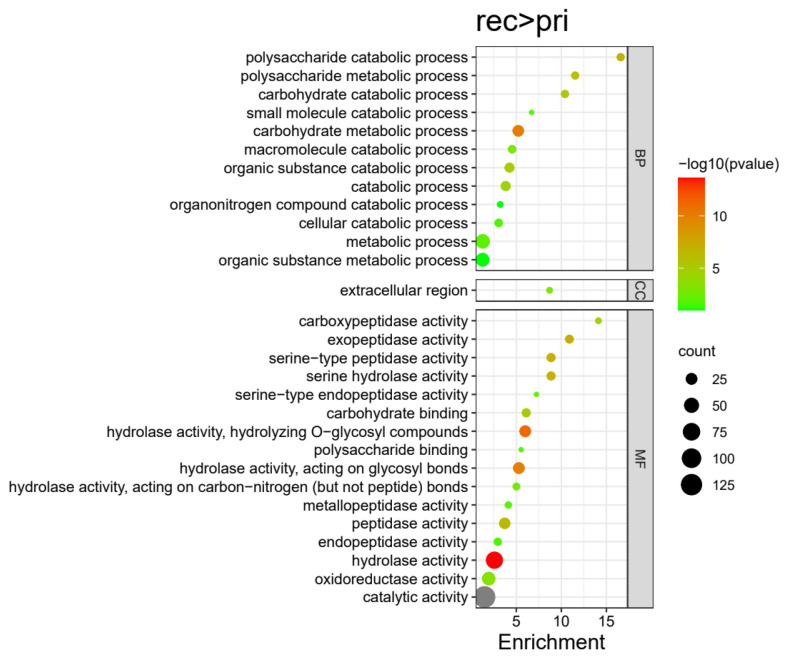
Enriched GO terms of highly expressed proteins in the Rec group compared with the Pri group in *H. marmoreus*. Bubble size indicates protein counts. Bubble color indicates p value. BP represents biological process, CC represents cellular component, and MF represents molecular function.

**Figure 9 jof-09-00225-f009:**
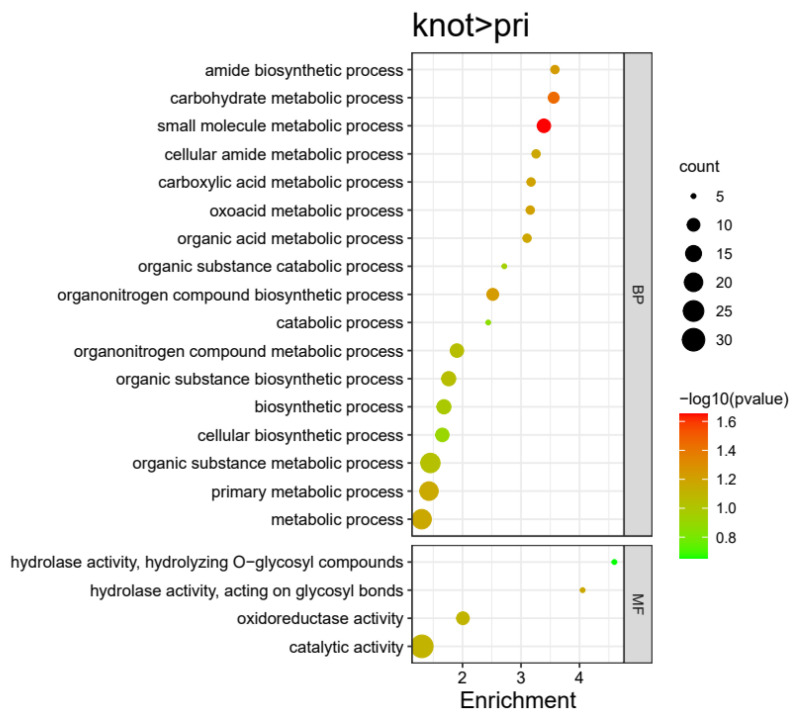
Enriched GO terms of highly expressed proteins in the Knot group compared with the Pri group in *H. marmoreus*. Bubble size indicates protein counts. Bubble color indicates p value. BP represents biological process, CC represents cellular component, and MF represents molecular function.

**Figure 10 jof-09-00225-f010:**
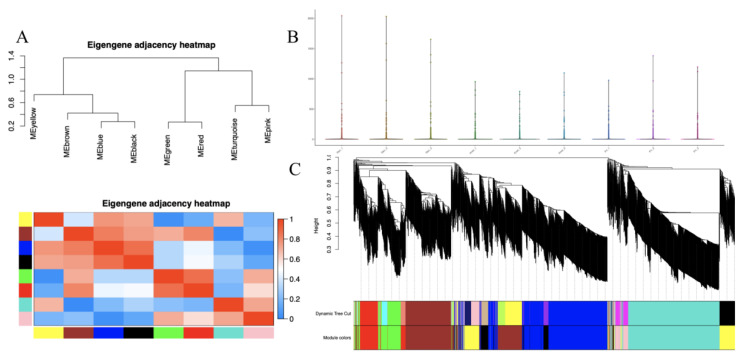
(**A**) Module correlation plot of the weighted gene co-expression network analysis. The eigengene is the first principal component gene of the specific module, representing the overall gene expression level of the module. (**B**) The sample distribution pattern of nine samples. (**C**) The module correlation plot of the weighted gene co-expression network analysis. Cluster genes by dissimilarity between genes, and the gene tree divided into eight different modules.

**Figure 11 jof-09-00225-f011:**
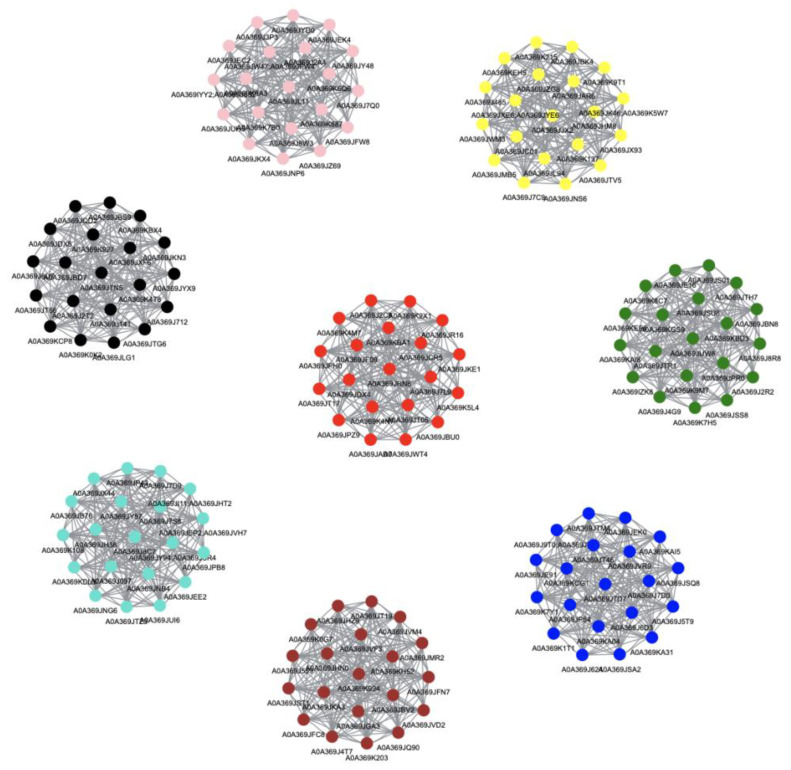
Proteins interactions with eight network visualizations of the WGCNA hub genes. Colors correspond to module colors, respectively.

**Table 1 jof-09-00225-t001:** Table of module eigengenes of the weighted gene co-expression network analysis.

Module	Rec_1	Rec_2	Rec_3	Knot_1	Knot_2	Knot_3	Pri_1	Pri_2	Pri_3
MEyellow	2.80 × 10^−1^	2.47 × 10^−1^	4.44 × 10^−2^	−4.33 × 10^−1^	−4.68 × 10^−1^	−4.76 × 10^−1^	3.42 × 10^−1^	2.19 × 10^−1^	2.44 × 10^−1^
MEbrown	−4.54 × 10^−1^	−4.63 × 10^−1^	−4.37 × 10^−1^	3.19 × 10^−1^	1.42 × 10^−1^	7.06 × 10^−5^	2.80 × 10^−1^	3.15 × 10^−1^	2.98 × 10^−1^
MEblue	−2.34 × 10^−1^	−2.41 × 10^−1^	−2.74 × 10^−1^	−1.81 × 10^−1^	−2.26 × 10^−1^	−2.49 × 10^−1^	5.12 × 10^−1^	5.00 × 10^−1^	3.93 × 10^−1^
MEblack	−1.54 × 10^−1^	−1.73 × 10^−1^	−1.82 × 10^−1^	−1.20 × 10^−1^	−2.00 × 10^−1^	−2.74 × 10^−1^	1.77 × 10^−1^	6.08 × 10^−2^	8.65 × 10^−1^
MEgreen	−2.79 × 10^−1^	−2.91 × 10^−1^	−2.65 × 10^−1^	4.07 × 10^−1^	6.03 × 10^−1^	3.72 × 10^−1^	−1.75 × 10^−1^	−1.97 × 10^−1^	−1.74 × 10^−1^
MEred	−3.04 × 10^−1^	−3.13 × 10^−1^	−2.60 × 10^−1^	8.28 × 10^−1^	7.81 × 10^−2^	1.78 × 10^−1^	−1.09 × 10^−1^	−7.61 × 10^−2^	−2.28 × 10^−2^
MEturquoise	5.18 × 10^−1^	4.85 × 10^−1^	4.03 × 10^−1^	−2.39 × 10^−1^	−1.99 × 10^−1^	−1.99 × 10^−1^	−2.53 × 10^−1^	−2.69 × 10^−1^	−2.48 × 10^−1^
MEpink	8.02 × 10^−2^	3.69 × 10^−2^	4.93 × 10^−1^	2.14 × 10^−1^	2.85 × 10^−1^	2.08 × 10^−1^	−4.56 × 10^−1^	−4.37 × 10^−1^	−4.25 × 10^−1^
MEgrey	−4.77 × 10^−1^	9.62 × 10^−2^	2.91 × 10^−1^	−1.02 × 10^−1^	−2.19 × 10^−1^	3.37 × 10^−1^	5.75 × 10^−1^	−9.32 × 10^−2^	−4.08 × 10^−1^

## Data Availability

The raw data for the proteomic analysis reported in this paper have been deposited in the OMIX, China National Center for Bioinformation/Beijing Institute of Genomics, Chinese Academy of Sciences (https://ngdc.cncb.ac.cn/omix (accessed on 1 February 2023): accession no. OMIX002711).

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
