# Peer review of "Comparative Proteomic Analyses within Three Developmental Stages of the Mushroom White Hypsizygus marmoreus"

_jof, 2023, doi:10.3390/jof9020225_

Round 1

Reviewer 1 Report

The work is interesting and it is performed according to traditional proteomic techniques. 

English details should be reviewed in some cases, like correlation between singular and plural adjectives after nouns.

Consistency in the use of capital letters (Rec or rec)

All scientific names should be written in italics

Some figures are too small and difficult to view (for example, fig 10)

I could not find reference 7 in the text.

Author Response

English details should be reviewed in some cases, like correlation between singular and plural adjectives after nouns.

Answer: We are really sorry for the confusion. Thank you for your suggestions. I already reviewed the English details of the manuscript.

Consistency in the use of capital letters (Rec or rec)

Answer: We are really sorry for the confusion. We really appreciate the reviewer’s critical and thoughtful comments on capital letters. I already make the use of capital letters (Rec, Pri, Knot) consistently.

All scientific names should be written in italics

Answer: We are really sorry for the confusion. We really appreciate the reviewer’s critical and thoughtful comments on italics. All scientific names are written in italics.

Some figures are too small and difficult to view (for example, fig 10)

Answer: We are really sorry for the confusion. We really appreciate the reviewer’s critical and thoughtful comments on Figure. I already changed a clear and bigger figure(fig 10).

I could not find reference 7 in the text.

Answer: Thank you for your suggestion. We have already adjusted the order of references of the manuscript. Reference 7 is in line 160 in the text: Afterward, enzymolysis was performed according to the FASP method created by Wis ́niewski et al. [27].

Reviewer 2 Report

There are three stages for the growth and development of Hypsizygus mamoreus. The transition stage is the key stage that mushrooms change from vegetable growth to reproductive stage. However the report about the detailed mechanism of the transition stage in Hypsizygus marmoreus is lack. The study provides important information for analyzing molecular mechanisms of fruiting body development.

However, the manuscript should be improved before publication.

Major comments:

(1) Key pathways or enzymes involved in H. marmoreus development should be listed in the Abstract parts, such as the enzymes or pathways discussed in the Discussion part.

 (2) The function of DEPs that could play key roles in H. marmoreus development in transition stage should compare with those reported in other mushrooms.

 (3) Careful proofread of this manuscript is needed, and the grammar of this manuscript also should be checked carefully.

 Minor comments:

(1) Line 22 “scratching to the thirteenth day after scratching are unclear” should be changed to “from scratching to primordium are unclear”.

 (2) Line 27 principal component (PC) analysis should be changed to “principal component analysis (PCA)

 (3) Line 36-44, Unclear, author should describe the key pathway in the development of H. marmoreus instead of list the DEPs.

 (4) Line 32-35, Line 49-52, Rec group should be used as control group, as in the Result part.

 (5) Line 70-75  The transition stage from vegetative growth to fruiting of Hypsizygus marmoreus can be divided into four stages: mycelial scratching stage, recovery growth stage, mycelial knotting stage and primordium stage (followed by the nail head stage).

 (6) Line 85-87, The water will be absorbed by the culture and activate the metabolic activity of the after-repening mycelia.

 (7) Line 128,143 etc.  Knot instead of Kink

 (8) Line 146 Proteomic analysis(12) Line 150, 165  UA or UT buffer?

 (9) Line 302, Figure 11A? The number of the Figure should listed from small to large.

(10) Line 308, 316. Figure 11?

 (11) Line 394 - 395 The biological characteristics for WCGNA analysis should be listed here or in the Methods part.

 (12) Line 463, to the thirteenth day or to the tenth day? Other parts should also checked.

 (13) Line 502 Reductase (which reductase?)

 (14) Line 518 add “identified ”

 (15) Line 554-555, “The recovery of mycelium was not affected by the location and area of scratching.” no evidence.

 (16) Line 545 551, the function of these proteins in the development of fungi in other papers should be discussed.

 (17) Line 571, the same problem, Rec should be used as control instead of the latter groups.
